# Treatment of refractory poly articular course juvenile idiopathic arthritis with tofacitinib: Extended experience from Bangladesh

Kazi Iman[1]*, Laboni Akter[2], Mohammad Masudur Rahman[3], Kamrul Laila[4], Mohammad Imnul Islam[4], Shahana A. Rahman[4]

1 Department of Pediatrics, Dr. M R Khan Shishu Hospital & ICH, Mirpur, Dhaka, Bangladesh, 2 Department of Pediatrics, Dhaka Medical College Hospital, Dhaka, Bangladesh, 3 Pediatrics, 250 Bedded General Hospital Jamalpur, Dhaka, Bangladesh, 4 Department of Pediatrics, Bangabandhu Sheikh Mujib Medical University, Shahbag, Dhaka, Bangladesh

* kaziiman28@gmail.com

## Abstract

### Background

Juvenile Idiopathic arthritis (JIA) is one of the most common chronic diseases in children. It still remains a challenge to treat refractory poly-articular course JIA patients, especially in Bangladesh, where patients from low socio-economic backgrounds are unable to manage biological agents. Tofacitinib is one of the alternative options to biological agents, which can be taken orally and is cost effective. The purpose of this prospective observational study was to evaluate the efficacy of tofacitinib in the treatment of refractory poly-articular course JIA cases.

### Materials and methods

This study was carried out in the Department of Pediatrics, Bangabandhu Sheikh Mujib Medical University (BSMMU). A total number of 50 refractory polyarticular course JIA patients received JAK-2 inhibitor, tofacitinib along with other drugs according to the recommended doses. The disease activity level was measured by Juvenile Arthritis Disease Activity Score-27 (JADAS-27). All the cases were assessed at baseline, 6th, 12th and52nd week of tofacitinib therapy. The relevant statistical tests were applied for data analysis.

### Results

After treating the refractory cases with tofacitinib, arthritis subsided, and laboratory parameters improved in all the cases. Overall JADAS-27 score improvement was 40.67%, 56.38% and 96% at 6th, 12th and 52nd week of follow-up respectively. It was also possible to taper the dose of steroid gradually and stopped it by 24 weeks. Tofacitinib was well tolerated with minimum side effects.

### Conclusions

Tofacitinib was effective to all the children with poly-articular course JIA. It was well tolerated and had very few tolerable adverse effects.

**Data Availability Statement:** All relevant data are within the manuscript and supporting information files.

**Funding:** The author(s) received no specific funding for this work.

**Competing interests:** The authors have declared that no competing interests exit.

## Introduction

Therapeutic interventions recommended for JIA in children includes non-steroidal anti-inflammatory drugs (NSAIDs), corticosteroids, non-biologic disease-modifying anti-rheumatic drugs (DMARDs) and biological DMARDs (bDMARDs) [1]. Patients who are taking NSAIDs and DMARDs may remain in clinical remission, but a subset of patients are unresponsive (refractory) to these agents. Biological DMARDs have good clinical outcomes but are very expensive. So, the treatments for refractory JIA patients remain a challenge in a developing country like Bangladesh.

JAK-inhibitors are effective new therapeutic approach for the treatment of JIA. Tofacitinib is one of the upright choices of JAK-inhibitors. It is the first-generation JAK-inhibitors, metabolized primarily in the liver via cytochrome P450 system [2, 3]. It has shown magnificent efficacy in the treatment of rheumatoid arthritis [4, 5]. Tofacitinib has both immune regulatory and anti-inflammatory characteristics which stop the JAK-signal transducers and activators of transcription (STAT) signaling pathway from being activated. This lessens the generation of pro-inflammatory cytokines and the inflammation-related harm brought on by immunological illnesses [6]. Evidences suggest, tofacitinib's potential role in the control of synovitis by altering innate and adaptive immune responses by preventing the production of interferon and interleukin-17 by human CD4 T cells [7, 8]. It has also been noted that this drug reduces the expression of chemokines in fibroblast-like synoviocytes that are stimulated by tumor necrosis factor [9]. After oral intake of tofacitinib, it is quickly absorbed and eliminated as it has a short half-life (3 hours only). Additionally, it has the benefit of minimal immunogenicity risk since it is not a monoclonal antibody [10]. Significant improvements in signs and symptoms, physical functioning, less disease flare-ups, and persistent clinical improvement were seen in the Phase 3 randomized double-blind placebo-controlled research [11]. On Sept 28, 2020, the US Food and Drug Administration (FDA) approved Pfizer's targeted synthetic drug tofacitinib for the treatment of polyarticular juvenile idiopathic arthritis [12].

A pilot study was carried out in our institute during January 2019 to September 2020 for 24 weeks, which found significant improvement of JADAS 27 in all the refractory poly-articular JIA cases after treatment with tofacitinib, and 70.4% cases had inactive disease at 24 weeks of follow-up [13].

The objective of the present study is to determine the efficacy of tofacitinib in the treatment of refractory cases of JIA after one year.

## Materials and methods

This was a prospective observational study conducted in the pediatric rheumatology clinic and inpatient Department of Pediatrics, Bangabandhu Sheikh Mujib Medical University (BSMMU), Dhaka, Bangladesh during October 2019 to June 2022. Ethical approval was taken from the institutional review board (IRB) of Bangabandhu Sheikh Mujib Medical University (BSMMU) (NO. BSMMU/2019/3937, Date: April 16, 2019).

### Participants

All diagnosed cases of refractory poly-articular course JIA patients (poly-articular including RF positive and negative, extended oligoarticular, enthesitis-related arthritis [ERA], systemic JIA without systemic features attending the pediatric rheumatology clinic and the inpatient Department of Pediatrics, BSMMU were enrolled in this study by purposive sampling. The present study included 15 cases from the previous pilot study [13]. After taking informed written consent, a predesigned questionnaire was completed for each patient by interviewing them or their parents. Relevant information was also collected from their medical records. Children

or parents unwilling to give consent, children with acute infection, chronic renal failure, liver failure and any lympho- proliferative disorders were excluded from this study. Other 12 cases enrolled in the previous study had irregular follow up, so they were excluded from the study. The sample size for this study was determined using the formula $z^2pq/d^2$, where z = 1.96 (at 95% confidence level); p = 0.06, as the prevalence or proportion of occurrence of JIA (Azam et al. 2012) [14]; and q = (1−p) = 0.94. Here, d represents absolute error and was set at 5%. Therefore, the sample size, calculated as n = $(1.96)^2 \times 0.06 \times 0.94/0.05^2$, was 86 patients. A total of 50 patients were enrolled in the study.

## Study design

JIA cases who failed to respond with methotrexate (MTX: subcutaneous route at a dose 15 mg/ $m^2$ body surface area) along with other adjuvant drugs over a period of 6 months were considered as refractory poly-articular course JIA patients [15]. Indications of adding tofacitinib was high disease activity despite adequate 1st line treatment along with adjunct therapy.

In this study, along with MTX, sulfasalazine (40 mg/kg/day) was used in ERA, leflunomide (10 mg/d) in poly-articular and extended oligoarticular and thalidomide (2 to 3 mg/kg/day) in systemic JIA cases were used as adjuvant drugs or co-medication. As in to relieve pain NSAIDs like indomethacin (2mg/kg/day) was given to ERA patients and naproxen (10mg/kg/day) was given to other cases respectively. Considering high disease activity level, children were on steroids at a starting dose of around 0.5mg/kg/day. All the JIA children received calcium and vitamin D combination along with folic/folinic acid routinely (1000 mg calcium and 400 vitamin D daily and folic/folinic acid 5 mg once weekly). After 6 months of MTX with adjuvant therapy, non-responders/minimum responders were considered as resistant cases. No biological DMARDS were used in this study.

## Procedure

After enrollment in the study, a detailed history was taken including age at presentation, age at diagnosis, disease duration, clinical presentation and treatment history including use of NSAIDS, DMARDs and steroids. A thorough physical examination was done including examination of tender joints, swollen joints, and limitation of movements, lymphadenopathy, skin rash, hepatomegaly, splenomegaly and evidence of serositis.

Baseline investigations included Hb%, total leukocyte count (TLC), differential count (DC), platelet count (PLT), erythrocyte sedimentation rate (ESR: normalized), serum alanine aminotransferase (ALT), serum creatinine, chest X- ray and routine and microscopic examination of urine. Normalized ESR was calculated using the formula [ESR (mm in 1st hour) −20] ÷10 [16]. Before starting tofacitinib, a repeat chest X-ray, Mantoux test and HBsAg test were done along with baseline investigations.

Disease activity was assessed by JADAS 27 score (Juvenile Arthritis Disease Activity Score in 27 joints) which included physician and patient global assessment of disease activity and acute- phase reactant (ESR) [16]. JADAS 27 scores higher than 8.5 were considered as high disease activity and less than 3.8 was considered as low disease activity [17]. Global assessments of disease activity by physician and patient/parent were measured using a visual analog scale (VAS).

Each case was treated with oral tofacitinib twice daily at a dose based on body weight. Children weighing >40 kg, weighing 25–40 kg and 15–25 kg were prescribed 5 mg, 4 mg and 3.5 mg twice daily respectively after the meal [5]. All the drugs were procured from a single pharmaceutical company. Along with tofacitinib, all the cases received MTX subcutaneously at a dose of 15 mg/$m^2$/wk. in a single weekly dose. Other drugs including steroids, other DMARDS

and NSAIDs were also continued. All the cases were followed up at the 6th, 12th and 52nd weeks to assess clinical status of disease activity using JADAS 27 and to identify any adverse effects.

In addition to clinical follow up, investigations including Hb%, TLC, DC, platelet, ESR, serum ALT, serum creatinine, urine routine and microscopic examinations were done at each follow ups. Base line data were compared with 6th, 12th and 52nd weeks, according to JADAS 27 criteria.

## Statistical analysis

In this study, a sample of 50 patients were analyzed using RStudio. For categorical variables, the data are presented as counts and percentages (n%). For continuous variables, the mean and standard deviation are provided. To determine if there are significant differences in the JADAS-27 variables between two consecutive weeks, a paired t-test was performed. The mean and standard deviation for each week. Additionally, changes in laboratory parameters were assessed using paired t-tests, with mean and standard deviation also reported for these results.

## Results

The age range of the cases at the onset of study was 5–17 years with a mean age of 13.6 ± 3.6 years. The mean age at disease onset was 9.6 ± 3.9 years. Disease duration was more than 3 years in the majority (61.8%) with a mean duration of 4.01 ± 2.16 years. Male: female ratio was 2:1 in this study. Thirty-two (62%) patients were taking steroids at the beginning of the study. Other adjuvants drugs were naproxen in 24 cases, indomethacin in 26 cases, sulfasalazine in 26 cases and leflunomide in 18 cases and thalidomide in 6 cases [Table 1].

Five types of JIA were observed in this study. Among them enthesitis related arthritis (ERA) was the most common (52%), followed by polyarticular RF negative JIA (18%), systemic JIA (12%), extended oligoarticular JIA (12%) and polyarticular RF positive JIA (6%). [Fig 1].

After administering tofacitinib (at prescribed doses), there was 42.06%, 61.38% and 92% improvement in the physician global assessment of VAS at 6 weeks, 12 weeks, and 52 weeks respectively. According to patient/parent global assessment, improvement was 42.74%, 52.3% and 94.48% at 6 weeks, 12 weeks and 52 weeks respectively. There were significant improvements of all other variables including active joint count and ESR. Overall improvement of JADAS-27 was 40.67%, 56.38%, and 96% at 6, 12 and 52 weeks respectively [Table 2].

From Table 3 it is found that, hemoglobin percentage of the cases significantly increased at all the follow-ups and ESR and platelet count significantly decreased at 12th and 52nd weeks

Table 1. Baseline demographic characteristics of cases including medication (n = 50).

| Characteristics | Mean ± SD |
|---|---|
| Age, year (at study onset) | 13.6±3.6 |
| Male: Female | 2:1 |
| Age, year (at disease onset) | 9.6±3.9 |
| Disease duration, year | 4.01±2.16 |
| Co-medication/adjuvant drugs | |
| Dose of Prednisolone at baseline, n = 32, mg/kg/d | 0.43 ± 0.25 |
| Dose of Naproxen, n = 24, mg/kg/d | 12.3 ± 1.1 |
| Dose of Indomethacin, n = 26, mg/kg/d | 2.10 ± 0.24 |
| Dose of Leflunomide, n = 18, mg/d | 9.9 ± 3.2 |
| Dose of Thalidomide, n = 6, mg/kg/day | 2.7±0.4 |
| Dose of Sulfasalazine, n = 26, mg/kg/d | 42.1 ± 1.37 |

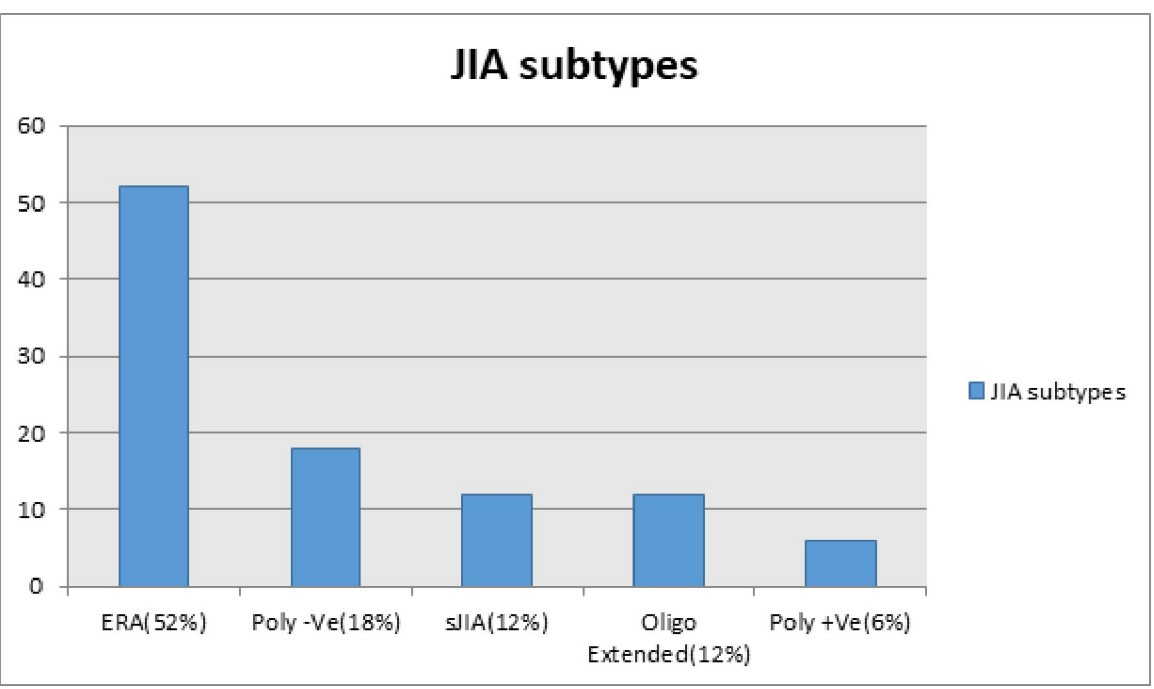

**Fig 1. The types of JIA among study group (n = 50).**

follow ups. Liver and renal functions were normal during this treatment period. Minor side effects like headache, vomiting and abdominal pain were observed initially, which subsided subsequently. Prednisolone dosages were significantly reduced from baseline to 6th and 12th weeks follow-up, and it was stopped at 24th week as shown in Table 4.

## Discussion

The Poly-articular course is the most aggressive form of JIA, where treatment is still challenging. The use of JAK-inhibitors is a new practice in pediatric rheumatology, and at present, there is limited and insufficient experience with JAK-inhibitors in JIA.

In this prospective observational study mean age at study enrollment was 13.6 year and mean age at disease onset was 9.6±3.9 year. Male: female ratio was 2:1. Presence of higher percentage of enthesitis related arthritis (53%) in this study might be the possible explanation of this very high male preponderance [Fig 1]. This finding was similar to two other studies conducted in the same center on JIA cases [18, 19] reflecting the presentation of JIA subtypes in a tertiary level hospital. In northern India, the picture is little different regarding age of onset

**Table 2. Changes in the JADAS-27 variable from baseline to follow up among study participant (n = 50).**

| Variables | At baseline | At 6 weeks | Improvement | p-value | At 12 weeks | Improvement | p- value | At 52 weeks | Improvement | p-value |
|---|---|---|---|---|---|---|---|---|---|---|
| Physician Global Assessment | 4.43 ± 1.44 | 2.59 ± 1.31 | 42.06 | 0.0001 | 1.00 ± 0.89 | 61.38 | 0.0001 | 0.08 ± 0.28 | 92.10 | 0.0001 |
| Patient Global Assessment | 5.32 ± 1.66 | 3.04 ± 1.14 | 42.74 | 0.0001 | 1.45 ± 1.14 | 52.3 | 0.0001 | 0.06 ± 0.27 | 94.48 | 0.0001 |
| Active joint count | 3.08 ± 1.27 | 1. ± 0.99 | 52.22 | 0.0001 | 0.265 ± 0.49 | 81.20 | 0.0001 | 0.11 ± 0.31 | 96.5 | 0.0001 |
| ESR | 4.91 ± 3.74 | 3.4 ± 3.79 | 30.75 | 0.028 | 1.83 ± 2.38 | 46.17 | 0.0001 | 0.46 ± 0.73 | 90.5 | 0.0001 |
| Total JADAS 27 | 17.7 ± 5.45 | 10.5 ± 5.46 | 40.67 | 0.0001 | 4.58 ± 3.99 | 56.38 | 0.0001 | 0.71 ± 1.16 | 96.0 | 0.0001 |

Abbreviation: ESR, erythrocyte sedimentation rete; JADAS 27, juvenile arthritis disease activity score in 27 joints. * p value was calculated by paired student's t test.

**Table 3. Changes in the laboratory parameters from baseline to follow up among study participants (n = 50).**

| Variables | At baseline | At 6 weeks p-value | At 12 weeks p-value | At 52 weeks p-value |
|---|---|---|---|---|
| Hb | 9.84±2.23 | 11.04±1.97 <0.001 | 11.44±1.89 <0.001 | 12.18±1.45 <0.001 |
| ESR | 66±38 | 50±42 0.0241 | 37±30 0.006 | 22.25±15 <0.004 |
| Total count | 11,257±3,241 | 11893± 0.1821 | 10398± 0.37 | 9810±2414 0.19 |
| | | 11693 | 3219 | |
| Neutrophil | 68.3±9.51 | 65.7±12 0.116 | 64.1±11 0.76 | 60.8±11.19 0.14 |
| Lymphocyte | 25.2±9.53 | 28±11 0.087 | 30.7±10.6 0.12 | 32.6±10.2 0.093 |
| Platelet | 449735±163237 | 435980± 0.613 | 382857± 0.001 | 320102± 0.001 |
| | | 121339 | 93719 | 79354 |
| Serum ALT | 23.8±21.5 | 17.8±11.6 0.04 | 18.4±7.14 0.835 | 16.5±5.99 0.09 |
| Serum creatinine | 0.49±0.14 | 2.43±9.31 0.04 | 0.54±0.20 0.16 | 0.60±0.19 0.05 |

*p value was calculated by paired student's t test

(6.7 ± 4.3 year), but similar about gender distribution 38 (67.8%) were boys and 18 (32.2%) were girls [20].

The prevalence of JIA subtypes varied widely across the geographic areas, with oligoarthritis being particularly prevalent in North America and Europe [21, 22]. In North Africa, systemic JIA is the most prevalent subtype, while in South East Asia, systemic JIA and enthesitis related arthritis patients are most prevalent [23]. In 2011 a cross-sectional study was conducted among school children aged 6–17 years in northern India. The estimated prevalence of JIA was 48/100,000 in Indian children [24]. Another study carried out from 1994 to 2006 in a community rheumatology clinic in Delhi, India showed very high prevalence of ERA (89% HLA-B27-positive) [25]. So, it may happen that, gender variation could depend on prevalence of particular JIA subtypes according to geographic and ethnic distribution. In the present study, ERA is the most common subtype, which is similar to previous studies done in the same country [18, 26].

In this study, disease activity was analyzed by JADAS 27, which is a reliable tool for determining disease activity states in JIA patients [16]. Mean Physican global assessment and patient global assessment significantly decreased at 6th, 12th and 52nd weeks of follow up [Table 2]. In 2023, a risk-benefit analysis was conducted to determine the viability of tofacitinib as a treatment for poly-articular course JIA, where it was found that tofacitinib was potentially effective in reducing flare-ups and lowering erythrocyte sedimentation rate (ESR) in immune-competent patients. Additionally, it was also found that tofacitinib was effective in patients who were refractory to traditional treatment [27].

The largest randomized, double-blind, placebo-controlled withdrawal study of tofacitinib in JIA showed that patients with polyarticular course of JIA treated with tofacitinib had significantly lower rate of disease flare at 44 weeks compared to placebo-group (29% vs 52.9%, p = 0.0031). ACR 30/50/70 response rates were also significantly higher in tofacitinib group (70.8% vs 47.1%, p = 0.003; 66.7% vs 47.1%, p = 0.017; 54.2% vs 37.1%, p = 0.039). The study concluded that tofacitinib is an effective treatment for JIA in pediatric populations [28].

**Table 4. Dose of steroids from baseline to follow up among study participants (n = 32).**

| Variable | At Baseline | At 6th week | p-value | At 12th week | p-value | At 24th week | p-value | At 52nd week | p-value |
|---|---|---|---|---|---|---|---|---|---|
| Dose of Prednisol1mg/kg/day | 0.52±0.26 | 0.26±0.16 | 0.000 | 0.03±0.04 | 0.000 | 0.00±0.00 | 0.000 | 0.00±0.00 | 0.000 |

*p value was calculated by paired student's t test

In the present study, after administration of tofacitinib, there was significant improvement of all the variables including patient global assessment, active joint count and ESR. Overall improvement of JADAS-27 was 40.67%, 56.38%, and 96% at 6, 12 and 52 weeks respectively [Table 2]. The previous pilot study from our center also showed significant improvement of all the parameters after introducing tofacitinib. Improvement of total score of JADAS 27 was also significant at 24th week follow up [13]. The present study followed up the cases up to 52nd week. The study done in Saint-Petersburg, Russia, showed similar picture, where corticosteroids could be tapered in 78.6% patients, median dose of corticosteroids was reduced from 0.25 to 0.1 mg/kg (p = 0.005) and stopped in 14.3% JIA patients [29]. A recent study on patients with Rheumatoid arthritis showed that, the rapid effect of tofacitinib in long standing RA patients allowed a significant reduction of the daily prednisolone dose leading to the discontinuation of glucocorticoids in up to 30% of patients, without limiting the drug effectiveness [30]. Similar result was found in the present study, where after adding tofacitinib, steroids were successfully tapered and discontinued by 24th week of follow up in all the cases. [Table 4]. The previous pilot study done in the same center also found similar result where dose of prednisolone was reduced significantly from baseline to 6 and 12 weeks follow up and at 24 weeks, steroid could be discontinued [13].

Hemoglobin level in the present study increased significantly at all the follow ups. The baseline investigations (including ESR and platelet count), were indicative of a high disease state which decreased significantly after treatment with tofacitinib [Table 3]. The pilot study conducted in our center showed similar results [13]. A study conducted in the USA showed that tofacitinib had the potential to be effective in reducing flare-ups and lowering ESR in immune-competent patients with poly articular course JIA [27].

Patients with refractory polyarticular course JIA, in this study showed a persistent clinical improvement at 52nd week after starting treatment with tofacitinib. The results of this trial may demonstrate a significant improvement in the management of a condition that has previously been shown to be challenging with conventional DMARDs. This study may therefore propose the use of tofacitinib under supervision in patients of refractory poly-articular course JIA.

## Conclusion

Tofacitinib can be used in all types of refractory cases of poly-articular course JIA patients. It significantly improved clinical symptoms and laboratory measures of disease activity in this trial. Additionally, expected tapering and ultimately withdrawal of steroid was also possible after adding tofacitinib. So, it can be concluded that tofacitinib has good efficacy in the treatment of refractory cases of JIA with minimum side effects. Additional multicenter prospective study may obtain a concrete recommendation.

## Supporting information

**S1 Dataset. Refractory polyarticular course of JIA treatment with tofacitinib.** All data are available in the attached file.
(XLSX)

## Acknowledgments

Authors are grateful to every patient who gave their consent for participation in this study; without their help it would be impossible to conduct this study.

## Author Contributions

**Data curation:** Kazi Iman, Laboni Akter, Mohammad Masudur Rahman, Kamrul Laila.

**Formal analysis:** Kamrul Laila.

**Methodology:** Mohammad Imnul Islam, Shahana A. Rahman.

**Software:** Kamrul Laila.

**Supervision:** Mohammad Imnul Islam, Shahana A. Rahman.

**Writing – original draft:** Kazi Iman.

**Writing – review & editing:** Kamrul Laila, Mohammad Imnul Islam, Shahana A. Rahman.

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
