## [Decision Letter · Decision Letter 0]

15 Aug 2024

PONE-D-24-20163Treatment of refractory poly articular course juvenile idiopathic arthritis with Tofacitinib: extended experience from BangladeshPLOS ONE

Dear Dr. Iman,

Thank you for submitting your manuscript to PLOS ONE. After careful consideration, we feel that it has merit but does not fully meet PLOS ONE’s publication criteria as it currently stands. Therefore, we invite you to submit a revised version of the manuscript that addresses the points raised during the review process.

We look forward to receiving your revised manuscript.

Kind regards,

Sadiq Umar

Academic Editor

PLOS ONE

Journal Requirements:

- https://www.frontiersin.org/articles/10.3389/fped.2022.820586/full

In your revision ensure you cite all your sources (including your own works), and quote or rephrase any duplicated text outside the methods section. Further consideration is dependent on these concerns being addressed.

3. We note that your Data Availability Statement is currently as follows: [All relevant data are within the manuscript and supporting information files.]

4. Please amend the manuscript submission data (via Edit Submission) to include authors Dr. Laboni Akter, Dr. Mohammad Masudur Rahman, Dr. Kamrul Laila, Dr. Mohammad Imnul Islam and Dr. Shahana A Rahman.

5. Please amend your list of authors on the manuscript to ensure that each author is linked to an affiliation. Authors’ affiliations should reflect the institution where the work was done (if authors moved subsequently, you can also list the new affiliation stating “current affiliation:….” as necessary).

7. Please include your tables as part of your main manuscript and remove the individual files. Please note that supplementary tables (should remain/ be uploaded) as separate "supporting information" files

Reviewers' comments:

Reviewer's Responses to Questions

**Comments to the Author**

1. Is the manuscript technically sound, and do the data support the conclusions?

Reviewer #1: No

Reviewer #2: Yes

2. Has the statistical analysis been performed appropriately and rigorously? 

Reviewer #1: No

Reviewer #2: No

3. Have the authors made all data underlying the findings in their manuscript fully available?

Reviewer #1: No

Reviewer #2: Yes

4. Is the manuscript presented in an intelligible fashion and written in standard English?

Reviewer #1: No

Reviewer #2: Yes

5. Review Comments to the Author

Reviewer #1: The manuscript titled "Treatment of Refractory Polyarticular Course Juvenile Idiopathic Arthritis with Tofacitinib: Extended Experience from Bangladesh" exhibits several critical issues that undermine its scientific credibility and readability.

1) Grammatical and Semantic Errors

The manuscript is plagued by numerous grammatical and semantic errors that detract from the overall readability and professionalism of the paper. This significantly hampers the reader's ability to follow the study's methodology, results, and conclusions.

2) Inconsistencies with Current Recommendations

The treatment approach described in the manuscript does not align with current clinical recommendations for juvenile idiopathic arthritis (JIA). According to standard guidelines, glucocorticoids should primarily be used as a bridging therapy, while NSAIDs and DMARDs remain the mainstay treatments for JIA. The manuscript fails to reflect this, suggesting a fundamental misunderstanding or misapplication of treatment protocols.

3) Omission of Intra-articular Steroid Use

The study does not mention whether intra-articular steroids were employed in the treatment regimen. This omission is significant, as intra-articular steroid injections are a common and effective treatment for JIA, especially for controlling localized inflammation.

4) Rationale for JAK Inhibitors Over bDMARDs

The manuscript asserts that the main rationale for opting for JAK inhibitors over biologic DMARDs (bDMARDs) is the cost. However, this rationale is flawed, as the prices of JAK inhibitors and bDMARDs are comparable. This discrepancy necessitates a more thorough explanation and justification within the manuscript.

5) Lack of Information on Side Effects

There is a glaring absence of information regarding the potential side effects of JAK inhibitors. Considering the known risks associated with these medications, including increased susceptibility to infections and other adverse effects, this omission is a significant oversight that compromises the study's thoroughness and reliability.

6) Inconsistent Inclusion Criteria

While the study purports to include all types of JIA, the conclusions focus solely on polyarticular JIA. This inconsistency raises questions about the study's design and the applicability of its findings to the broader JIA population.

7) Poorly Constructed Tables

The tables presented in the manuscript are poorly written, lacking necessary units and clarity. It is not evident from the tables how many patients received different types of treatment and for how long, which is crucial for interpreting the study's results.

8) Overall Assessment

The manuscript contains fundamental errors and deficiencies that severely compromise its scientific validity and utility. The numerous grammatical and semantic errors, inconsistencies with current treatment recommendations, lack of detailed methodology, and omission of critical information about side effects and treatment specifics render the manuscript unsuitable for publication. Given these issues, I recommend rejection without the possibility of revision.

In conclusion, this manuscript fails to meet the necessary standards for scientific publication and does not contribute reliable or valuable insights into the treatment of juvenile idiopathic arthritis with tofacitinib.

Reviewer #2: The manuscript suggests the use of tofacitinib in juvenile arthritis. This manuscript is scientifically sound but requires minor revisions. It is acceptable upon completion of these revisions.

Comments:

Comment: In abstract line 28: I would suggest to change, “country like ours” to “especially in Bangladesh”.

Comment: Authors should mention that tofacitinib is FDA approved drug to treat moderate to severe rheumatoid arthritis.

Comment: Line 146 and 178 authors described the percentage of ERA 53% and RF negative JIA 17% which does not match with the figure 1. Please elaborate this point.

Comment: The authors showed the no of patients (n=50). Weather these 50 patiesnts tested in all of the variables showed in table 2? For example all 50 patients were tested for Physician Global Assessment, Patient Global Assessment, Active joint count, ESR and Total JADAS? Or few were tested for Physician Global Assessment and few Patient Global Assessment and so on.

Comment: The authors should mention what kind of statistics they used and how they did it? Such as the softwere name or manual calculation in Microsoft excel.

6. PLOS authors have the option to publish the peer review history of their article (what does this mean?). If published, this will include your full peer review and any attached files.

Reviewer #1: No

Reviewer #2: **Yes: **Mohd Tayyab

---

## [Author Response · Author response to Decision Letter 0]

2 Sep 2024

Dear Reviewers,

Thank you reviewers for our generous review. We have edited our article according to your guideline. We believe

now this manuscript is suitable for publication.

---

## [Editor Report · Decision Letter 1]

9 Sep 2024

PONE-D-24-20163R1Treatment of refractory poly articular course juvenile idiopathic arthritis with Tofacitinib: extended experience from BangladeshPLOS ONE

Dear Dr. Iman,

Thank you for submitting your manuscript to PLOS ONE. After careful consideration, we feel that it has merit but does not fully meet PLOS ONE’s publication criteria as it currently stands. Therefore, we invite you to submit a revised version of the manuscript that addresses the points raised during the review process.

**ACADEMIC EDITOR: **The manuscript presents interesting findings, but it would benefit greatly from professional English editing to enhance clarity, grammar, and overall readability. Improving the language will ensure that the key messages and data are communicated effectively to a broader audience. I recommend revising the manuscript to address issues with sentence structure, word choice, and punctuation to make the text more polished and precise.Please submit your revised manuscript by Oct 24 2024 11:59PM. If you will need more time than this to complete your revisions, please reply to this message or contact the journal office at plosone@plos.org. Please include the following items when submitting your revised manuscript:A rebuttal letter that responds to each point raised by the academic editor and reviewer(s). You should upload this letter as a separate file labeled 'Response to Reviewers'.A marked-up copy of your manuscript that highlights changes made to the original version. You should upload this as a separate file labeled 'Revised Manuscript with Track Changes'.An unmarked version of your revised paper without tracked changes. You should upload this as a separate file labeled 'Manuscript'.If applicable, we recommend that you deposit your laboratory protocols in protocols.io to enhance the reproducibility of your results. Protocols.io assigns your protocol its own identifier (DOI) so that it can be cited independently in the future. For instructions see: https://journals.plos.org/plosone/s/submission-guidelines#loc-laboratory-protocols. Additionally, PLOS ONE offers an option for publishing peer-reviewed Lab Protocol articles, which describe protocols hosted on protocols.io. Read more information on sharing protocols at https://plos.org/protocols?utm_medium=editorial-email&utm_source=authorletters&utm_campaign=protocols.

We look forward to receiving your revised manuscript.

Kind regards,

Sadiq Umar

Academic Editor

PLOS ONE

Journal Requirements:

Additional Editor Comments:

The manuscript presents interesting findings, but it would benefit greatly from professional English editing to enhance clarity, grammar, and overall readability. Improving the language will ensure that the key messages and data are communicated effectively to a broader audience. I recommend revising the manuscript to address issues with sentence structure, word choice, and punctuation to make the text more polished and precise.

---

## [Author Response · Author response to Decision Letter 1]

18 Sep 2024

Dear reviewers,

Thank you for your generous comments on the manuscript. We have edited our manuscript according to your guideline. Changing areas are marked with yellow colour.

We hope that manuscript is suitable for publication.

Here, I have attached Data sheet, Title page with author list.

---

## [Editor Report · Decision Letter 2]

3 Oct 2024

Treatment of refractory poly articular course juvenile idiopathic arthritis with Tofacitinib: extended experience from Bangladesh

PONE-D-24-20163R2

Dear Dr. Iman,

We’re pleased to inform you that your manuscript has been judged scientifically suitable for publication and will be formally accepted for publication once it meets all outstanding technical requirements.

Kind regards,

Sadiq Umar

Academic Editor

PLOS ONE

---

## [Editor Report · Acceptance letter]

22 Oct 2024

PONE-D-24-20163R2 

PLOS ONE

Dear Dr. Iman, 

I'm pleased to inform you that your manuscript has been deemed suitable for publication in PLOS ONE. Congratulations! Your manuscript is now being handed over to our production team.

Kind regards, 

on behalf of

Dr. Sadiq Umar 

Academic Editor

PLOS ONE